# Effect of upper limb isometric training (ULIT) on hamstring strength in early postoperative anterior cruciate ligament reconstruction patients: Study protocol for a randomized controlled trial

Efri Noor Muhamad Hendri[1]☯, Mohamad Shariff A. Hamid[2]*,
Badrul Akmal Hisham Md. Yusoff[1,3]☯, Norlelawati Mohamad[1]☯, Ashril Yusof[4]☯

1 Department of Orthopedics and Traumatology, Hospital Canselor Tuanku Muhriz, National University of Malaysia, Kuala Lumpur, Malaysia, 2 Department of Sports Medicine, Faculty of Medicine, Universiti Malaya, Kuala Lumpur, Malaysia, 3 Department of Orthopaedics and Traumatology, Faculty of Medicine, Universiti Kebangsaan Malaysia, Cheras, Kuala Lumpur, Malaysia, 4 Faculty of Sports and Exercise Science, Universiti Malaya, Kuala Lumpur, Malaysia

☯ These authors contributed equally to this work.
* ayip@um.edu.my

## Abstract

Anterior cruciate ligament (ACL) injuries impact approximately 68.6 per 100,000 individuals annually, with ACL reconstruction (ACLR) being a common intervention for restoring knee stability in physically active individuals. Despite advancements in surgical techniques and rehabilitation protocols, patients often experience prolonged recovery, hamstring weakness, and neuromuscular deficits, increasing the risk of re-injury and osteoarthritis. Early-phase ACLR rehabilitation primarily focuses on managing pain, swelling, and quadriceps strength, frequently neglecting the critical role of hamstrings in knee stabilization. This leaves a gap in addressing imbalances that hinder functional recovery and return-to-sport timelines. Upper limb isometric training (ULIT) presents an innovative approach to enhance hamstring activation during the early rehabilitation phase. By leveraging the posterior myofascial kinetic chain (PMKC), ULIT indirectly stimulates hamstrings through bilateral static upper limb exercises, such as wall push up, shoulder extension and scapular retraction, promoting neuromuscular coordination and kinetic chain synergy. These exercises mitigate challenges associated with direct hamstring loading, such as arthrogenic muscle inhibition and graft protection needs. Preliminary research suggests upper limb resistance exercise at submaximal voluntary contraction facilitates inter-limb strength gains, improves core abdominals and hamstring activation, and reduces knee imbalances, supporting accelerated recovery and reduced re-injury risk. The ULIT demonstrates potential as an alternative warm-up exercise to promote hamstring activation and enhance overall readiness for physical activity. Emerging

**Data availability statement:** No datasets were generated or analyzed during the current study. All relevant data from this study will be made available upon study completion.

**Funding:** The author(s) received no specific funding for this work.

**Competing interests:** The authors have declared that no competing interests exist.

findings highlight ULIT as a safe and potentially effective supplementary intervention, but further research is essential to establish its role in ACLR rehabilitation and develop evidence-based protocols. This study aims to evaluate the effects of integrating ULIT into standard care rehabilitation on hamstring strength and physical function in early-phase postoperative ACLR patients with hamstring autograft. The findings could introduce a novel and effective strategy to optimize recovery, enhance functional outcomes, and support a safer return to sport.

**Trial registration number:** ACTRN12624001445561 and available at https://www.anzctr.org.au/Trial/Registration/TrialReview.aspx?id=388441&isReview=true.

## Introduction

Anterior cruciate ligament (ACL) injuries affect 68.6 per 100,000 individuals annually [1], with ACL reconstruction (ACLR) recommended for physically active individuals experiencing knee instability [2]. Globally, ACLRs have surpassed 100,000 per year, growing annually by 2% [3]. Despite advancements, ACLR patients face prolonged recovery, with physical function limitations and an inability to return to sports for at least six months [4]. Factors such as graft type (e.g., hamstring autografts) and rehabilitation protocols contribute to chronic hamstring weakness and neuromuscular deficits, increasing the risk of re-injury, altered biomechanics, and osteoarthritis [5–7].

Current ACLR rehabilitation programs focus on managing pain and swelling, restoring quadriceps strength, and achieving full knee extension during the early postoperative phase, which spans the first 12 weeks [8,9]. Although these objectives are essential, they often overshadow the importance of hamstring rehabilitation. Rehabilitation protocols frequently emphasize quadriceps-dominant strategies, leaving a critical gap in addressing the hamstrings' role in knee stabilization, hip and torso positioning, and injury prevention. Furthermore, early rehabilitation is hindered by challenges such as arthrogenic muscle inhibition (AMI), which disrupts neural pathways and impairs quadriceps and hamstring activation [10,11]. The necessity to protect the graft from excessive tensile forces further limits direct hamstring loading and stretching during this period [12].

Consequently, hamstring recovery within current protocols is often delayed, leading to imbalances in knee musculature that hinder overall functional recovery [13]. Structured neuromuscular rehabilitation has demonstrated improvements in muscle strength, but few clinical practice guidelines (CPGs) specifically address early hamstring strengthening, despite its vital role in reducing ACL shear forces and stabilizing the knee joint during dynamic activities [14,15]. High imbalances in knee muscle strength during early rehabilitation phases also slow recovery in the final stages [16,17].

Innovative approaches that integrate indirect hamstring activation techniques, such as Upper Limb Isometric Training (ULIT), offer promising solutions. ULIT leverages the posterior myofascial kinetic chain (PMKC) to indirectly enhance core abdominal muscles and hamstring activation through upper limb isometric exercises [18,19].

Biomechanical models suggest that force transmission through the PMKC occurs via fascial connections between the latissimus dorsi, thoracolumbar fascia, gluteus maximus, and hamstrings, facilitating neuromuscular coordination and interconnected muscle synergies [20–23]. Experimental evidence, particularly electromyography (EMG) studies, supports this mechanism, demonstrating significant enhancements in core abdominal and hamstring activation during upper limb isometric contractions [24–27]. EMG analysis reveals that shoulder abduction and extension at 50% MVC elicit a 15–25% increase in biceps femoris activation compared to resting conditions, while scapular retraction exercises augment semitendinosus activity by approximately 10–18% [24,28]. This indirect activation is attributed to kinetic chain dynamics, which optimize neuromuscular recruitment, improve joint stability, and enhance movement efficiency. By addressing knee imbalances and addressing challenges such as pain, swelling, and graft protection requirements in the early stages of anterior cruciate ligament reconstruction (ACLR) rehabilitation, ULIT offers a safe and effective strategy for improving hamstring strength and stability. Incorporating ULIT into early ACLR rehabilitation programs may accelerate recovery and mitigate the risk of reinjury.

Research indicates that isometric training at 50% of maximum voluntary contraction can promote inter-limb strength gains, making ULIT a promising early rehabilitation tool [29–31]. It prepares patients for dynamic movements, reduces knee imbalances, and supports functional recovery. Incorporating ULIT into early ACLR rehabilitation programs could optimize hamstring strength and stability, accelerate recovery and reduce the risk of re-injury.

While the theoretical framework and preliminary research provide insights into the potential mechanisms, physiological adaptations, and clinical implications of upper limb isometric training (ULIT) for hamstring strength during early-phase ACLR rehabilitation, robust clinical evidence remains limited. Current evidence offers limited guidance on the optimal intensity, volume, and frequency of ULIT, leaving its impact on hamstring strength and functional outcomes in the ACLR population less established [32–35].

This study aims to investigate the effects of incorporating specific bilateral ULIT into standard rehabilitation care on hamstring strength and physical function in early-phase postoperative ACLR patients with hamstring autograft.

We hypothesize that participants receiving ULIT in addition to standard care will demonstrate better hamstring strength at 12 weeks post-operatively compared to those receiving standard care rehabilitation alone. Emerging findings suggest ULIT as a safe and potentially effective supplementary therapy for enhancing hamstring activation. However, further research is needed to clarify its clinical role and develop evidence-based protocols to improve patient outcomes.

The primary objective of this study is to investigate the effects of ULIT combined with standard care rehabilitation on hamstring strength in early phase postoperative ACLR patients. The secondary objectives are to compare the effects of ULIT and standard care on improving physical function, enhancing hamstring flexibility, and assessing patient adherence post-ACLR.

## Materials and methods

### Study design

This study is a longitudinal, parallel-group, concealed allocation, assessor-blinded, randomised (1:1) controlled trial conducted at a tertiary hospital. Participants who meet the eligibility criteria, provide informed consent, and complete baseline measurement testing will be randomly assigned to the study. Participants will be informed that they will be randomly allocated to one of two study groups: 1) the intervention group (ULIT combined with the standard care ACLR rehabilitation protocol) or 2) the control group (standard care rehabilitation protocol).

### Ethics

The trial will be conducted at the Orthopaedic Clinic and Physiotherapy Unit of Hospital Canselor Tuanku Muhriz (HCTM), Kuala Lumpur. Ethics approval for the trial protocol (version 1, 24 September 2024) was obtained from the Institutional Review Board (IRB) of the Universiti Kebangsaan Malaysia Research Ethics Committee (RECUKM)(Ethics Ref. no: JEP-2024–860) (S1 Appendix). The protocol has been registered with the Australian and New Zealand

Clinical Trials Registry (http://www.anzctr.org.au) (Registration no: ACTRN12624001445561 and available at https://www.anzctr.org.au/Trial/Registration/TrialReview.aspx?id=388441&isReview=true.), ensuring compliance with the ethical principles outlined in the Declaration of Helsinki (2000).

## Sample size

Sample size calculations were conducted using IBM SPSS Statistics (Version 29) for the primary outcome: differences between the means of hamstring maximum voluntary isometric contraction peak force. Based on the population standard deviation of 0.71 for the primary outcome measure, as reported in the previous study [11], and setting the Type I error at 0.05, the Type II error at 0.2, and the power at 80%, the minimum required sample size is 14 patients per group. To account for an expected attrition rate of 10%, 16 participants are planned for inclusion in each study group.

## Participants recruitment

Eligible participants for this study are patients diagnosed with an ACL tear, with or without a concomitant meniscal injury, who are scheduled for ACL reconstruction surgery. Prospective participants will receive an invitation to join the study at least two weeks prior to their surgery. They will be identified from the elective surgery list at the Orthopaedic Clinic of HCTM. Eligibility of participants will be determined through the patient's medical records, a physical examination conducted by the surgeon, and direct communication with the patient. The principal investigator will explain the details of the trial to potential participants and provide them with written information for reference (S2 Appendix). Participants are required to provide written consent (S3 Appendix) to confirm their participation. Patients who choose not to participate will maintain standard rehabilitative care.

## Inclusion and exclusion criteria

Participants eligible for inclusion must be aged between 18 and 45 years and scheduled for ACLR using an ipsilateral hamstring tendon autograft, with or without a meniscal injury. They must also be proficient in English and capable of providing informed consent. Participants will be excluded if they are undergoing revision ACL surgery, have multi-ligament injuries such as instability in the collateral, posterolateral, medial, or posterior cruciate ligaments, are using allografts, or have sustained upper limb or contralateral lower limb injuries after ACL surgery.

## Randomization

Following baseline assessments, participants will be randomly allocated to either the intervention or control group through a computer-generated sequence with a 1:1 allocation ratio. Group assignments will be hidden within sealed, numbered envelopes, which will be securely stored in a locked cabinet. Non-research personnel from the Orthopaedic Department will unseal the next envelope and notify the attending physiotherapist of the group allocation.

## Blinding

Outcome assessors and data managers will remain blinded to treatment allocation. Participants will be instructed not to disclose their treatment received to the physician or medical officer performing assessments. Treatment allocation will only be revealed in exceptional cases, such as potential harm or critical care needs. Upon the conclusion of the trial, physiotherapists and patients were requested to speculate on the treatment administered to each participant. The success of blinding will be assessed using the 'blinding index' as described by James et al. [36]. Fig 1 depicts the SPIRIT schedule and an overview of the study design.

| | STUDY PERIOD | | | | |
|---|---|---|---|---|---|
| | Enrolment | Allocation | Post-allocation | | |
| **TIMEPOINT** | **-t₁** | **T₀ (Pre-operative)** | **T₁ (4 week)** | **T₂ (8 week)** | **T₃ (12 week)** |
| **ENROLMENT** | X | | | | |
| Eligibility screen | X | | | | |
| Informed consent | | X | | | |
| Allocation | | X | | | |
| **INTERVENTIONS** | | | | | |
| ULIT + Standard ACLR | | ←——————————————————→ | | | |
| Standard ACLR | | ←——————————————————→ | | | |
| **ASSESSMENTS** | | | | | |
| Quadriceps and hamstring strength | | X | X | X | X |
| IKDC-SKE form | | X | | | X |
| Active knee extension test | | X | X | X | X |
| Adherence | | ←——————————————————→ | | | |
| Adverse event | | ←——————————————————→ | | | |

*IKDC-SKE: International Knee Documentation Committee Subjective Knee, ULIT: Upper Limb Isometric Training, ACLR: Anterior Cruciate Ligament Rehabilitation*

**Fig 1. SPIRIT schedule of enrollment, interventions, and assessments.**

## Study outcomes

Patients' sociodemographic information, including gender, date of birth, ethnic background, sport and level of participation, will be recorded. Additionally, information on education level, physical activity level, and previous medical history will be collected.

**Primary outcome: quadriceps and hamstring maximum voluntary isometric strength (MVIC).** The primary outcome of this study is the peak force of quadriceps and hamstring muscles strength assessed using the maximum voluntary isometric contraction (MVIC) tests. The MVIC test will be performed at enrolment (pre-operative), 4-, 8- and 12-weeks follow-up sessions. The maximal voluntary isometric contraction (MVIC) will be measured using a built-in handheld dynamometer and inclinometer (Active Force 2™), following the methodology described by a previous

study [14]. Patients will be positioned in high sitting with hips flexed at 90° and knees bent at 60°. The dynamometer will be securely placed 5 cm above the lateral malleolus to ensure consistent force measurement. Participants will be instructed to exert maximal effort during the contraction while maintaining proper positioning and stabilization. This standardized approach, as outlined by previous studies, ensures reliable and reproducible measurements of MVIC in assessing hamstring strength. Patients will perform three MVICs of knee extension, each lasting five seconds, to evaluate quadriceps strength, followed by knee flexion testing for the hamstrings [37]. A one-minute rest period will be provided between quadriceps and hamstring tests to reduce muscle fatigue [38]. This standardized approach ensures reliable and reproducible measurements of MVIC for both muscle groups.

The assessor will provide consistent instructions and verbal encouragement during tests. Participants will be instructed to exert maximum force by pushing with their quadriceps or pulling with their hamstrings against the dynamometer. Both the average and best peak force scores will be recorded [39].

**Secondary outcomes: International Knee Documentation Committee Subjective Knee Form (IKDC-SKF) questionnaire.** Patients will complete the IKDC-SKF questionnaire (S4 Appendix) to evaluate their perception of daily knee function and associated symptoms [7,40]. The questionnaire is straightforward, allowing patients to understand and complete it independently. It has been specifically designed and validated for patients undergoing ACL reconstruction, and it covers all the domains in the International Classification of Functioning, Disability, and Health (ICF) [41]. The IKDC has good validity and responsiveness, with excellent test-retest reliability 84% confirmation of preset hypotheses for construct validity and 86% for responsiveness [40,42].

**Secondary outcomes: Active knee extension test (AKET).** Active knee extension test (AKET) for hamstring range of movement assessment. The AKE test is an active test that is safe as the participant dictates the endpoint and has been recommended and often used to measure hamstring tightness [43]. The AKET will be performed using an inclinometer to measure the knee extension angle (KEA). The participant lies supine, with their hip in a 90° flexion position and a flat position for the contralateral leg. Patients will be asked to extend their knee until they feel maximum tightness in the posterior thigh without pain, maintaining a relaxed ankle position to reduce the impact of the gastrocnemius muscle. One or more of the following three criteria can determine the endpoint for AKET: (a) the examiner's perception of firm resistance; (b) the visible onset of pelvic rotation; and (c) the participant experiencing a strong but tolerable stretch, slightly before the onset of pain. In a previous study [44], researchers proposed a clinically significant improvement of 10.2°.

The AKET has demonstrated high reliability in evaluating hamstring flexibility in healthy people, with intraclass correlation coefficients (ICC) of 0.87 for the dominant knee and 0.81 for the non-dominant knee, while the intra-rater (test-retest) reliability ICC scores varied between 0.75 and 0.97 [45].

## Interventions

**Intervention group: upper limb isometric training combined with standard ACLR rehabilitation program.** Participants in the intervention group will perform ULIT in addition to the standard ACLR rehabilitation program from weeks 4–12 post-surgery. A trained physiotherapist will administer the ULIT program, providing verbal instructions, demonstrations, and written guidelines (S5 Appendix).

ULIT Protocol:

The ULIT program consists of three isometric exercises performed in a standing position (S5 Appendix):

1. Wall push-ups: Participants stand facing a solid wall, feet shoulder-width apart, approximately arm's length from the wall. With hands positioned at shoulder height, they perform an isometric push-up by applying force against the wall while maintaining arm alignment. Each push-up is held for 5 seconds, followed by 5 seconds of rest. Participants perform 1 set of 5 repetitions, once or twice daily.

2. Shoulder extension: Participants position themselves with their backs against the wall, arms fully extended downward at their sides. They push their arms backward against the wall while maintaining a neutral spine and full knee extension. Each repetition is held for 5 seconds, followed by 5 seconds of rest. One set of 5 repetitions is performed, once or twice daily.

3. Shoulder external rotation: Participants position themselves sideways to the wall, with their elbow flexed at a 90-degree angle and tucked against their side. They press the back of their hand against the wall, engaging the shoulder muscles without changing arm position. Each repetition is held for 5 seconds, followed by 5 seconds of rest. One set of 5 repetitions is performed, once or twice daily.

Intensity and Neuromuscular Considerations:

To maximize neuromuscular activation while minimizing fatigue, patients will perform these exercises at 50% of their maximum voluntary contraction (MVC). Research indicates that intensities at or above 40% MVC effectively stimulate neuromuscular adaptations without causing excessive fatigue [28]. Specifically, training at 50% MVC optimally recruits Type IIA muscle fibers, which are critical for enhancing kinetic chain efficiency while limiting early fatigue and excessive muscle tension [46,47].

Lower intensities, such as 30% MVC, primarily activate slow-twitch fibers, resulting in limited force transmission and reduced neuromuscular adaptation. Conversely, higher intensities such as 70% MVC are more likely to induce early fatigue and compromise mechanical efficiency, potentially hindering sustained participation in rehabilitation exercises [48,49].

Implementing 50% MVC in the ULIT protocol achieves an optimal balance, effectively enhancing neuromuscular activation while ensuring safety and practicality in ACLR rehabilitation.

Furthermore, Xu et al. highlight the importance of optimizing muscle activation and movement strategies for both injury prevention and rehabilitation, supporting our intensity selection [50].

General instructions:

Effort Level: The first repetition of each exercise is performed at maximal effort (100%), whereas the subsequent five repetitions are performed at 50% effort.

Rest Intervals: A 1-minute interval is advised between exercises.

Environment: Exercises should be performed against a stable wall (e.g., concrete or brick) on a non-slip floor surface to ensure stability and safety.

Safety and Modifications:

Participants are instructed to maintain steady breathing throughout the exercises and avoid breath-holding. If maintaining the correct posture is challenging, they may reduce intensity by stepping closer to the wall. Participants are advised to stop the exercise if they experience sharp pain, excessive fatigue, or difficulty maintaining controlled breathing.

In addition to ULIT, patients in the intervention group will perform the standard care ACL rehabilitation protocol (S6 Appendix).

**Control group: standard ACLR rehabilitation program.** Patients in the control group will receive the standard ACLR rehabilitation program from the Department of Medical Rehabilitation Services (HCTM). Five physiotherapists, each with over five years of experience in musculoskeletal rehabilitation, will oversee the sessions. The standard ACLR rehabilitation program includes knee mobilization exercises, cryotherapy, progressive resistance exercises (e.g., eccentric strengthening), proprioceptive training, and dynamic stabilization drills. Each follow-up session lasts approximately one hour, and patients will be instructed to perform these exercises at home 1–2 sessions per day for up to three months post-surgery. Additionally, all participants are encouraged to adhere diligently to their prescribed daily home exercise program.

## Adherence and adverse events monitoring

All patients are required to document their home-based exercises and report any adverse events using an exercise diary provided in the form of a weekly Google Form. Participants will be classified as adhered (≥3 sessions/week) or non-adhered (≤2 sessions/week) based on their average weekly exercise frequency [51]. A physiotherapist will conduct weekly follow-ups via phone or WhatsApp to support adherence. The study will set an adherence target of a self-perceived rating ≥80% [52].

Adverse events will be documented using a non-leading questionnaire as part of participants' training diaries throughout the intervention (S7 Appendix). Participants may contact the PI or physiotherapist at any time during the study. All adverse events, regardless of outcome, will be reported and published. A summary of this trial is outlined in Fig 2.

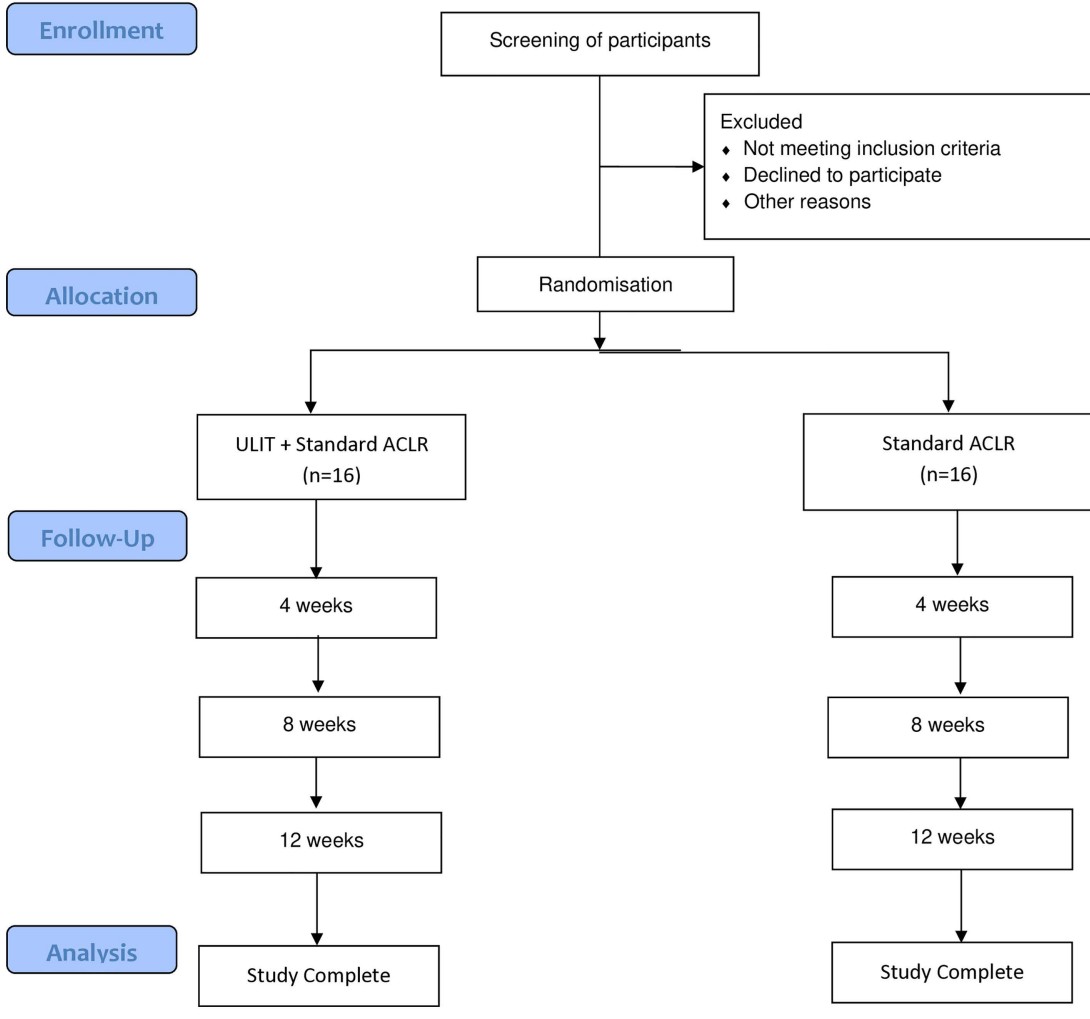

**Fig 2. Study flowchart.**

## Data management

Access to the data will be restricted to the supervisor, Principal Investigator (PI), co-researchers, and independent statistician, with no disclosure to unauthorized parties. The supervisor and PI will advise trial investigators on monitoring withdrawals, ensuring ethical conduct, addressing missing data, and reviewing major adverse events. A case report form (CRF) will document unavailable data to ensure consistency, accuracy, and proper analysis. All data will be securely stored in a locked cabinet in the PI's office and retained for at least five years following study completion and publication.

## Statistical analysis

Data analysis will be conducted using the SPSS Windows Version 29.0 software (SPSS, Chicago, IL, USA). Baseline characteristics of the intervention and control groups will be presented to assess group equivalence and identify potential confounders. Continuous data will be expressed as mean±standard deviation (SD), or median ± interquartile range (IQR) based on data distribution, while categorical data will be presented as frequency (n) and percentage (%). Comparisons between groups will be performed using independent t-tests for continuous variables and chi-square tests for categorical variables. Any significant differences between groups will be acknowledged and adjusted for in the final analysis to ensure accurate outcome interpretation.

For the primary outcome variable, the principle of intention-to-treat (ITT) will be applied, including data from participants with incomplete data or protocol deviations. To address missing data, we will assess the missing data mechanism, missing completely at random (MCAR), missing at random (MAR), or missing not at random (MNAR) using diagnostic tests such as Little's MCAR test to determine the suitability of multiple imputation (MI). If the data align with MAR or MCAR, MI will be used; however, if an MNAR pattern is detected, alternative strategies such as sensitivity analyses or pattern-mixture models will be explored to ensure the robustness of our findings.

If the missing rate exceeds 20%, additional sensitivity analyses will be performed using alternative approaches including case analysis and last observation carried forward (LOCF) to assess the robustness of the findings [53,54].

One-way repeated-measures ANOVA will be used to explore changes in hamstring and quadriceps MVIC within and between groups over time. A p-value of <0.05 would be significant.

## Rigour

The study protocol has been developed in conformity with the evidence-based guidelines of the SPIRIT 2013 Statement [55] on the essential elements of a clinical trial protocol. The study design conforms to the CONSORT 2010 guidelines for conducting parallel group randomized controlled trials (RCTs). The TIDieR checklist is added as a tool to describe and replicate the intervention proposal (S8 Appendix). Upon completion of the study, the data collected here will be made available to those investigators who request it.

## Status of study

Participant recruitment will commence in January 2025 and is expected to conclude by January 2026. The intervention phase and final follow-up assessments are anticipated to be completed by May 2026.

## Discussion

This study will focus on evaluating the effects of ULIT on hamstring strength deficits in individuals after undergoing ACLR. The primary outcome, MVIC of hamstring and quadriceps peak force, will be measured pre-operatively and at 4-, 8-, and 12-weeks post-operatively. Secondary outcomes will assess hamstring flexibility, knee symptoms, physical activity levels, and knee-related quality of life. Previous research has demonstrated significant correlations between patient-reported and objective outcomes, providing insight into muscle function deficits associated with ACLR [9,12]. ULIT is believed to

have potential as an early post-operative ACLR rehabilitation, complementing standard ACLR rehabilitation programs by enhancing knee strength, flexibility, and functional outcomes.

This study's strength lies in its focused design, which targets individuals within a specific age range who have undergone ACLR with a hamstring graft. This approach ensures greater uniformity in participant characteristics, thus enhancing internal validity. The study effectively captures essential early recovery milestones by focusing on the critical early rehabilitation phase, spanning from pre-operation to three months post-surgery. The standardized eight-week ULIT intervention further strengthens consistency across participants, while the inclusion of a control group receiving standard ACLR care allows for a robust comparison and clearer evaluation of the intervention's effectiveness.

However, this study has some limitations. One notable limitation is the absence of a cost-effectiveness analysis, which was not feasible given the modest sample size, despite its recognized importance. Additionally, while our study is designed to assess the short-term impact of ULIT, we acknowledge the need to explore potential compensatory loads and imbalances that may arise from prolonged ULIT use beyond 12 weeks.

To address this, future research should incorporate biomechanical assessments to monitor long-term neuromuscular adaptations. Recommended approaches include surface electromyography (sEMG) to evaluate muscle activation patterns and firing rates, along with video kinematic analysis to assess movement quality and detect compensatory strategies. These assessments will provide valuable insights into the potential risks of upper limb overuse and core stability imbalances, ultimately guiding the refinement of ULIT protocols to ensure both effectiveness and long-term safety in ACLR rehabilitation.

Future studies with extended follow-up periods exceeding 6–9 months are also warranted to assess the sustained benefits of ULIT in achieving positive clinical outcomes and minimizing injury risk during the return-to-sport phase.

## Conclusion and clinical relevance

This randomized clinical trial will evaluate the effect of upper limb isometric training on hamstring strength in ACLR patients with hamstring autografts. No previous trials have examined upper limb resistance training combined with standardized early-stage ACLR rehabilitation. If effective, this approach could enhance current treatment strategies and introduce a novel rehabilitation method.

## Supporting information

**S1 Appendix.  Study Protocol for Ethics Application.**
(PDF)

**S2 Appendix.  Participant Information Sheets.**
(PDF)

**S3 Appendix.  Participant Consent Form.**
(PDF)

**S4 Appendix.  International Knee Documentation Committee Subjective Knee Form (IKDC-SKF) Questionnaire.**
(PDF)

**S5 Appendix.  Upper Limb Isometric Training (ULIT) Protocol.**
(PDF)

**S6 Appendix.  Anterior Cruciate Ligament Reconstruction (ACLR) Exercise Protocol.**
(PDF)

**S7 Appendix.  Adverse Events Form.**
(PDF)

## S8 Appendix. TIDieR Checklist.
(PDF)

## S9 Appendix. Study Ethics Approval.
(PDF)

## S10 Appendix. SPIRIT Checklist.
(PDF)

## Acknowledgments

The authors would like to express their gratitude to all the patients who will participate in the study, as well as to Hospital Canselor Tuanku Muhriz and Universiti Malaya for their invaluable support.

## Author contributions

**Conceptualization:** Efri Noor Muhamad Hendri, Mohamad Shariff A. Hamid, Badrul Akmal Hisham Md. Yusoff, Ashril Yusoff.

**Data curation:** Norlelawati Mohamad.

**Investigation:** Mohamad Shariff A. Hamid, Norlelawati Mohamad.

**Methodology:** Efri Noor Muhamad Hendri, Mohamad Shariff A. Hamid, Badrul Akmal Hisham Md. Yusoff, Norlelawati Mohamad, Ashril Yusoff.

**Resources:** Mohamad Shariff A. Hamid.

**Supervision:** Mohamad Shariff A. Hamid, Ashril Yusoff.

**Validation:** Mohamad Shariff A. Hamid.

**Writing – original draft:** Efri Noor Muhamad Hendri, Mohamad Shariff A. Hamid, Badrul Akmal Hisham Md. Yusoff, Norlelawati Mohamad, Ashril Yusoff.

**Writing – review & editing:** Efri Noor Muhamad Hendri, Mohamad Shariff A. Hamid, Badrul Akmal Hisham Md. Yusoff, Norlelawati Mohamad, Ashril Yusoff.

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
