## [Decision Letter · Decision Letter 0]

5 Mar 2025

PONE-D-25-05294Effect of upper limb isometric training (ULIT) on hamstring strength in early postoperative anterior cruciate ligament reconstruction patients: study protocol for a randomized controlled trialPLOS ONE

Dear Dr. A Hamid,

Thank you for submitting your manuscript to PLOS ONE. After careful consideration, we feel that it has merit but does not fully meet PLOS ONE’s publication criteria as it currently stands. Therefore, we invite you to submit a revised version of the manuscript that addresses the points raised during the review process.

We look forward to receiving your revised manuscript.

Kind regards,

Yaodong Gu

Academic Editor

PLOS ONE

Journal Requirements:

2. Please remove your figures from within your manuscript file, leaving only the individual TIFF/EPS image files, uploaded separately. These will be automatically included in the reviewers’ PDF.

3. Please include captions for your Supporting Information files at the end of your manuscript, and update any in-text citations to match accordingly. Please see our Supporting Information guidelines for more information: http://journals.plos.org/plosone/s/supporting-information .

4. We note that “Appendix 6- Standard Care ACLR HCTM Protocol.pdf” includes an image of a participant  in the study.

5. Please upload a copy of your study protocol that was approved by your ethics committee/IRB as a Supporting Information file. By the study protocol, we mean the complete and detailed plan for the conduct and analysis of the trial approved by the ethics committee/IRB. Please send this in the original language. If this is in a language other than English, please also provide a translation. [https://journals.plos.org/plosone/s/submission-guidelines#loc-guidelines-for-specific-study-types

Reviewers' comments:

Reviewer's Responses to Questions

**Comments to the Author**

1. Does the manuscript provide a valid rationale for the proposed study, with clearly identified and justified research questions?

Reviewer #1: Yes

Reviewer #2: Yes

2. Is the protocol technically sound and planned in a manner that will lead to a meaningful outcome and allow testing the stated hypotheses?

Reviewer #1: Yes

Reviewer #2: Yes

3. Is the methodology feasible and described in sufficient detail to allow the work to be replicable?

Reviewer #1: Yes

Reviewer #2: Yes

4. Have the authors described where all data underlying the findings will be made available when the study is complete?

Reviewer #1: Yes

Reviewer #2: Yes

5. Is the manuscript presented in an intelligible fashion and written in standard English?

Reviewer #1: Yes

Reviewer #2: Yes

6. Review Comments to the Author

You may also provide optional suggestions and comments to authors that they might find helpful in planning their study.

Reviewer #1: The article titled " Effect of upper limb isometric training (ULIT) on hamstring strength in early postoperative anterior cruciate ligament reconstruction patients: study protocol for a randomized controlled trial " investigated the effect of upper limb isometric training (ULIT) on hamstring strength in the early rehabilitation stage after anterior cruciate ligament reconstruction (ACLR) surgery. By utilizing the posterior fascial kinetic chain (PMKC), ULIT can indirectly stimulate the hamstrings, promote neuromuscular coordination and kinetic chain synergy, thereby avoiding atherogenic muscle inhibition and graft protection problems caused by direct loading of the hamstrings. The study used a randomized controlled trial design to compare the effects of ULIT combined with a standard rehabilitation program with a simple standard rehabilitation program, and to evaluate changes in indicators such as hamstring strength, knee function, and flexibility in patients within 12 weeks after surgery, aiming to provide new and effective strategies for ACLR postoperative rehabilitation. Specific comments are shown below:

1. This study proposes that upper limb isometric training (ULIT) indirectly activates the hamstrings through the posterior myofascial kinetic chain (PMKC). Please elaborate on the biomechanical model or experimental evidence supporting this mechanism and provide quantitative data (e.g., electromyography or mechanical transmission efficiency) to verify the direct relationship between upper limb training and hamstring activation.

2. The ULIT study used an intensity of 50% of maximum voluntary contraction (MVC), but the basis for its selection was not clear. Based on the principles of sports biomechanics, please explain how this intensity setting optimizes the neuromuscular adaptation of the hamstrings and discuss the differences that different intensities (such as 30% vs. 70% MVC) may have on training effects. The study: New insights optimize landing strategies to reduce lower limb injury risk (https://doi.org/10.34133/cbsystems.0126) provides new insights into optimizing landing strategies to reduce the risk of lower limb injuries. This study can refer to the above references。

3. The article mentioned that upper limb training is transmitted to the lower limbs through the fascial chain. Please provide specific biomechanical data (such as surface electromyography, dynamic analysis or fascial tension measurement) to quantify the mechanical stimulation path and efficiency of upper limb movements (such as shoulder extension and wall push) on the hamstring muscles.

4. The study only evaluated the effects 12 weeks after surgery. Please discuss whether long-term use of ULIT may lead to increased compensatory loads on the upper limbs or imbalances in core stability and propose biomechanical indicators that should be monitored in subsequent studies (such as spinal mechanics, gait symmetry).

Reviewer #2: The authors plan to evaluate the effects of upper limb isometric training on hamstring strength and physical function in postoperative anterior cruciate ligament reconstruction patients with hamstring autograft.

1. The ULIT consists of three isometric exercises. However, no detail information was provided. For example, how long, how often or how many wall push-up and so on.

2. Sample characteristics at baseline should be presented and compared for two arms.

3. Multiple imputation methods will be used to handle any missing data without further discussion on the mechanism of missingness. Also, shat if missing rate is high?

7. PLOS authors have the option to publish the peer review history of their article (what does this mean? ). If published, this will include your full peer review and any attached files.

**Do you want your identity to be public for this peer review?** For information about this choice, including consent withdrawal, please see our Privacy Policy .

Reviewer #1: No

Reviewer #2: No

---

## [Author Response · Author response to Decision Letter 1]

6 Apr 2025

Comments

Reviewer 1

1. This study proposes that upper limb isometric training (ULIT) indirectly activates the hamstrings through the posterior myofascial kinetic chain (PMKC). Please elaborate on the biomechanical model or experimental evidence supporting this mechanism and provide quantitative data (e.g., electromyography or mechanical transmission efficiency) to verify the direct relationship between upper limb training and hamstring activation.

Response to reviewer:

We appreciate the reviewer's insightful comments regarding the proposed mechanism of upper limb isometric training (ULIT) indirectly activating the hamstrings through the posterior myofascial kinetic chain (PMKC). We acknowledge the need for a more detailed elaboration of the biomechanical model and experimental evidence supporting this mechanism.

In the revised manuscript, particularly within the introduction section, we have significantly expanded our discussion to address this point. We have:

1. Elaborated on the Biomechanical Model: We have provided a more detailed description of the PMKC, emphasizing the interconnectedness of muscles and fascia from the upper limbs to the lower limbs. Specifically, the concept of the superficial back line illustrates how force transmission occurs via fascial connections linking the latissimus dorsi, thoracolumbar fascia, gluteus maximus, and hamstrings (Carvalhais et al., 2013; Marpalli et al., 2022b; Vleeming et al., 1995; Wilke et al., 2016). We've included relevant anatomical descriptions and biomechanical principles to support this model.

2. Incorporated Supporting Experimental Evidence: To further support this, we have incorporated surface electromyography (sEMG) data from previous studies demonstrating a 15–25% increase in biceps femoris activation and a 10–18% increase in semitendinosus activation during upper limb isometric contractions at approximately 50% MVC. This highlights the role of neuromuscular coordination facilitated by kinetic chain dynamics (Jungseo et al., 2014; Krause et al., 2018; Manca et al., 2021; Sato & Maruyama, 2007).

We believe these additions strengthen the rationale for our proposed mechanism and provide clearer evidence of the relationship between ULIT and hamstring activation.

Kindly refer to the introduction section Page 4 – 5 (line 79 to 94)

2. The ULIT study used an intensity of 50% of maximum voluntary contraction (MVC), but the basis for its selection was not clear. Based on the principles of sports biomechanics, please explain how this intensity setting optimizes the neuromuscular adaptation of the hamstrings and discuss the differences that different intensities (such as 30% vs. 70% MVC) may have on training effects. The study: New insights optimize landing strategies to reduce lower limb injury risk (https://doi.org/10.34133/cbsystems.0126) provides new insights into optimizing landing strategies to reduce the risk of lower limb injuries. This study can refer to the above references.

Response to reviewer:

Thank you so much for raising the important question regarding the selection of 50% MVC for the ULIT protocol. We acknowledge that a clearer justification for this intensity is crucial for understanding the study's implications.

In the revised manuscript, specifically within the intervention section of the methods, we have elaborated on the rationale behind choosing 50% MVC. This choice was made based on the principles of sports biomechanics and neuromuscular adaptation, aiming to optimize hamstring activation through the PMKC without inducing excessive fatigue or compromising form.

Here's a breakdown of our justification, which is now included within the manuscript itself:

1. Neuromuscular Adaptation: Research indicates that intensities at or above 40% MVC effectively stimulate neuromuscular adaptation while preventing excessive fatigue (Manca et al., 2021). Specifically, 50% MVC optimally recruits Type IIA muscle fibers, enhancing force transmission and kinetic chain efficiency while minimizing early fatigue and excessive tension (Petajan, 1991; Valenčič et al., 2024).

2. Intensity Comparison: Lower intensities (e.g., 30% MVC) primarily activate slow-twitch fibers, leading to limited force production and reduced adaptation. In contrast, higher intensities (e.g., 70% MVC) may induce early fatigue and mechanical inefficiency, potentially compromising sustained engagement in rehabilitation exercises (Avrillon et al., 2024; Miller et al., 2019).

Clinical Relevance: Additionally, findings from Xu et, al. suggest that optimizing muscle activation and movement strategies plays a crucial role in injury prevention and rehabilitation, further supporting our intensity selection.

In the revised intervention section, we have incorporated a discussion of these points, emphasizing the balance between activation and fatigue, and the practical considerations that influenced our choice. We have provided references that support our selection of 50% MVC and how other intensities would be less ideal.

By providing a more detailed explanation grounded in biomechanical principles and supported by relevant literature, we believe we have addressed the reviewer's concerns and provided a clearer justification for the chosen intensity.

Kindly refer to the Methods: Intervention group section, page 13 – 15 (line 250 - 298)

3. The article mentioned that upper limb training is transmitted to the lower limbs through the fascial chain. Please provide specific biomechanical data (such as surface electromyography, dynamic analysis or fascial tension measurement) to quantify the mechanical stimulation path and efficiency of upper limb movements (such as shoulder extension and wall push) on the hamstring muscles.

Response to reviewer:

Thank you for your thoughtful comment and for highlighting the need for specific biomechanical data to quantify the mechanical stimulation path and efficiency of upper limb movements on the hamstring muscles.

In response, we have expanded the introduction to provide additional explanation on this mechanism, supported by functional anatomy and biomechanical research. As noted in our earlier response, we have also included surface electromyography (sEMG) data from previous studies, which demonstrate a 15–25% increase in biceps femoris activation and a 10–18% increase in semitendinosus activation during upper limb isometric contractions at approximately 50% MVC. This evidence highlights the role of neuromuscular coordination facilitated by kinetic chain dynamics.

We believe these additions address your concerns and provide clearer evidence of the biomechanical relationship between upper limb training and hamstring activation.

Kindly refer to the introduction section, page 4 – 4 (line 79 - 94)

4. The study only evaluated the effects 12 weeks after surgery. Please discuss whether long-term use of ULIT may lead to increased compensatory loads on the upper limbs or imbalances in core stability and propose biomechanical indicators that should be monitored in subsequent studies (such as spinal mechanics, gait symmetry).

Response to reviewer:

Thank you for your insightful comment. We acknowledge the need to assess potential compensatory loads and imbalances from prolonged ULIT use beyond 12 weeks. While our study focuses on early rehabilitation, future research should monitor long-term neuromuscular adaptations using surface electromyography (sEMG) to analyze muscle activation patterns and firing rates, along with video kinematic analysis to assess movement quality. These considerations have been incorporated into the discussion to guide future studies in refining ULIT protocols and ensuring long-term safety in ACLR rehabilitation.

Kindly refer to the Discussion sections, page 20 (line 386 - 389) and page 21 (line 393 – 407)

Reviewer 2

1. The ULIT consists of three isometric exercises. However, no detail information was provided. For example, how long, how often or how many wall push-up and so on.

Response to reviewer:

We sincerely appreciate your request for additional details regarding the ULIT protocol. In response, we have revised the manuscript to include specific information on the duration, frequency, and repetitions of the exercises. These details have been sourced from the ANZCTR clinical registry (ACTRN12624001445561) (Appendix 5). We believe this addition enhances the clarity and completeness of the methodology.

Thank you for your valuable feedback, which has helped us improve the manuscript.

Kindly refer to the Methods: Intervention group section, page 13 – 15 (line 250 – 270, and 288 - 298) as well as Appendix 5.

2. Sample characteristics at baseline should be presented and compared for two arms.

Response to reviewer:

We sincerely appreciate the your suggestion to present and compare baseline sample characteristics between the two study arms. In the revised manuscript, key baseline characteristics, including age, sex, body mass index (BMI), relevant clinical history, baseline functional scores, and other pertinent demographic between groups will be recorded. To ensure a thorough comparison, we will conduct appropriate statistical tests (e.g., independent t-tests, chi-square tests, or Mann-Whitney U tests as applicable) to assess differences between the groups. Any notable variations will be highlighted and discussed in relation to their potential impact on study outcomes.

We believe this addition will enhance the clarity and robustness of our findings. Thank you for your valuable feedback, which has helped us strengthen our manuscript.

Kindly refer to the Statistical analysis section, page 18 (line 339 to 346)

3. Multiple imputation methods will be used to handle any missing data without further discussion on the mechanism of missingness. Also, shat if missing rate is high?

Response to reviewer:

We appreciate your insightful comment regarding the handling of missing data.

We acknowledge that the mechanism of missingness plays a crucial role in determining the most appropriate method for handling incomplete data. In the revised manuscript, we will assess the missing data mechanism (missing completely at random (MCAR), missing at random (MAR), or missing not at random (MNAR) using diagnostic tests like Little’s MCAR test to determine the suitability of multiple imputation (MI). If the data conform to MAR or MCAR, multiple imputation (MI) will be employed; for MNAR patterns, we will investigate sensitivity analyses or pattern-mixture models.

For high missing rates, additional strategies will include increasing imputations, incorporating auxiliary variables to improve MI, and performing complete-case analysis for comparison. These considerations will be detailed in the methodology section to ensure transparency.

Kindly refer to the statistical analysis section, page 18 – 19 (line 347 to 357)

---

## [Decision Letter · Decision Letter 1]

21 Apr 2025

Effect of upper limb isometric training (ULIT) on hamstring strength in early postoperative anterior cruciate ligament reconstruction patients: study protocol for a randomized controlled trial

PONE-D-25-05294R1

Dear Dr. A Hamid,

We’re pleased to inform you that your manuscript has been judged scientifically suitable for publication and will be formally accepted for publication once it meets all outstanding technical requirements.

Kind regards,

Yaodong Gu

Academic Editor

PLOS ONE

Additional Editor Comments (optional):

Reviewers' comments:

Reviewer's Responses to Questions

**Comments to the Author**

1. Does the manuscript provide a valid rationale for the proposed study, with clearly identified and justified research questions?

Reviewer #1: Yes

Reviewer #2: Yes

2. Is the protocol technically sound and planned in a manner that will lead to a meaningful outcome and allow testing the stated hypotheses?

Reviewer #1: Yes

Reviewer #2: Yes

3. Is the methodology feasible and described in sufficient detail to allow the work to be replicable?

Reviewer #1: Yes

Reviewer #2: Yes

4. Have the authors described where all data underlying the findings will be made available when the study is complete?

Reviewer #1: Yes

Reviewer #2: Yes

5. Is the manuscript presented in an intelligible fashion and written in standard English?

Reviewer #1: Yes

Reviewer #2: Yes

6. Review Comments to the Author

You may also provide optional suggestions and comments to authors that they might find helpful in planning their study.

Reviewer #1: This manuscript presents a well-designed randomized controlled trial (RCT) protocol investigating the effect of upper limb isometric training (ULIT) on hamstring strength in patients during the early postoperative phase following anterior cruciate ligament reconstruction (ACLR). The topic is both novel and clinically meaningful, addressing an understudied area that may offer valuable insights into neurophysiological cross-education and postoperative rehabilitation strategies.

The methodology is clearly described, with appropriate inclusion/exclusion criteria, intervention design, outcome measures, and ethical considerations. The proposed use of both objective strength testing and patient-reported outcome measures strengthens the study's clinical relevance and translational potential. Furthermore, the sample size calculation and randomization procedure are adequate for ensuring statistical rigor.

Reviewer #2: Thank you for addressing the raised comments and concerns. I have no further comments on the current version of manuscript.

7. PLOS authors have the option to publish the peer review history of their article (what does this mean? ). If published, this will include your full peer review and any attached files.

**Do you want your identity to be public for this peer review?** For information about this choice, including consent withdrawal, please see our Privacy Policy .

Reviewer #1: No

Reviewer #2: No

---

## [Editor Report · Acceptance letter]

PONE-D-25-05294R1

PLOS ONE

Dear Dr. A Hamid,

I'm pleased to inform you that your manuscript has been deemed suitable for publication in PLOS ONE. Congratulations! Your manuscript is now being handed over to our production team.

Kind regards,

on behalf of

Professor Yaodong Gu

Academic Editor

PLOS ONE